# Crosstalk Classification Based on Synthetically Consider Crosstalk and Fragmentation RMCSA in Multi-Core Fiber-Based EONs

**Yanbo Chen** [1,2]**, Nan Feng** [3]**, Yue Zhou** [1,2]**, Danping Ren** [1,2] **and Jijun Zhao** [1,2,*]

[1] School of Information and Electrical Engineering, Hebei University of Engineering, Handan 056038, China
[2] Hebei Key Laboratory of Security and Protection Information Sensing and Processing, Handan 056038, China
[3] Hebei Key Laboratory of Photonic Information Technology and Application, the 54th Research Institute of CETC, Shijiazhuang 050081, China
* Correspondence: zjijun@hebeu.edu.cn

**Abstract:** Space division multiplexing elastic optical networks (SDM-EONs) based on multi-core fiber (MCF) technologies have attracted widespread attention as a potential means of enhancing large capacity and high flexibility. However, inter-core crosstalk (XT) degrades the quality of transmission. The algorithm for minimizing XT leads to an increase in spectrum fragmentation in the lightpath, which influences the spectrum utilization. Therefore, the question of how to comprehensively consider the two factors and improve the network performance is an issue worthy of study. This paper focuses on maximizing spectrum resource utilization while satisfying the XT constraints. Firstly, we optimize a three-dimensional metric model to evaluate XT and fragmentation more exactly in SDM-EONs. Furthermore, a crosstalk classification (CC) algorithm, which can adjust the XT constraints according to the actual situation of the network, is proposed. Moreover, to match the CC algorithm, we describe the crosstalk and fragmentation in the network and propose a synthetically consider crosstalk and fragmentation (SCCF) algorithm. Finally, simulation results show that the proposed CC-SCCF routing, modulation, core, and spectrum allocation algorithm reduces the XT on existing lightpaths, and also provides a lower probability of blocking and greater spectrum utilization.

**Keywords:** space division multiplexing (SDM); elastic optical networks (EONs); routing; modulation; core; and spectrum assignment (RMCSA); fragmentation; inter-core crosstalk (XT)

## 1. Introduction

In recent years, with the rapid development of network applications and the increase in bandwidth-hungry requests, internet traffic has shown exponential growth, which has led to a huge challenge for the capacity and efficiency of optical networks [1]. The networks which are based on wavelength division multiplexing cannot meet the current network demand. Elastic optical networks (EONs) have become an effective technology to solve the network capacity problem due to their flexible and efficient characteristics [2]. By dividing the spectrum into multiple frequency slots (FSs), EONs can flexibly select paths, modulation formats, and spectrum resources for lightpaths. At the same time, the current capacity of single-core fiber technology is close to the Shannon capacity limit [3]. Space division multiplexing (SDM), based on multi-core fiber (MCF) applications, can effectively expand optical fiber capacity by utilizing spatial dimensions [4]. Thus, SDM-EONs, which combine MCF-based SDM and EONs, have become one of the effective solutions with which to overcome the capacity crisis [5].

In EONs, since the arrival and departure of requests are random, dynamic routing, modulation, and spectrum allocation (RMSA) algorithms are usually adopted when allocating resources to requests. In this process, the algorithm needs to meet three constraints, namely spectrum continuity, contiguity, and non-overlap [6]. However, with the dynamic

establishment and release of services in the network, some idle spectrum resources become unavailable because they do not meet the requirements of spectrum continuity and spectrum adjacency. The idle but unavailable spectrum resources are called spectrum fragments. With the increasing frequency spectrum fragments, frequency utilization decreases, blocking probability increases and network performance decreases [5]. Therefore, in order to improve network performance, the impact of fragmentation on the network should be taken into account when allocating resources.

In SDM-EONs based on MCF, the introduction of a spatial dimension expands RMSA into routing, modulation, core, and spectrum allocation (RMCSA), posing more severe challenges in terms of resource allocation [5,6]. At the same time, services on the same frequency gap between adjacent fibers affect each other, resulting in inter-core crosstalk (XT), affecting the quality of transmission (QoT) and degrading network performance [7]. Therefore, crosstalk is also a factor to be taken into account during resource allocation.

Based on the above content, both the crosstalk problem and the fragmentation problem will affect the overall performance of the network. Meanwhile, considering the XT-fragmentation problem can comprehensively improve the transmission quality and spectrum utilization [8]. Note that reducing the impact of XTS on the network leads to service fragmentation, which increases the number of free spectrum blocks, increases resource fragmentation and decreases resource utilization [9,10]. On the contrary, when the number of fragments decreases, the service distribution becomes compact and the XT of adjacent cores increases, affecting QoT [11]. The over-optimization of one leads to the degradation of the other. Therefore, we believe that, in the process of resource allocation, we should fully consider how to improve the spectrum utilization rate under the conditions of meeting the XT constraint.

In order to reduce XT and fragmentation in networks, several resource management algorithms have been proposed. The first consideration is the development of typical resource management algorithms, which reroute all or part of the requested lightpath in the network to reduce XT and fragmentation. However, this will lead to traffic interruption. On this basis, the interruption-free algorithm is proposed, but this has the disadvantages of a lack of complex optical elements and high extra equipment cost [12]. To avoid the above problems, a fragmentation awareness algorithm and an XT awareness algorithm have been proposed and have demonstrated the capacity to improve spectrum utilization and reduce XT without creating any traffic disruption in EONs.

Therefore, we propose a sensing algorithm that considers both XT and fragmentation. Lightpaths can be established and transmitted on the network only when requests meet the crosstalk constraint. Therefore, our algorithm will first consider meeting the XT threshold to ensure the quality of transmission, and then further consider the comprehensive impact of the XT issue and fragmentation issue on the request. First, we extend the existing three-dimensional crosstalk model, and obtain the crosstalk-fragmentation metric model with three domains. Secondly, in order to better satisfy the XT constraint, according to the actual situation of the network, a crosstalk classification (CC) algorithm is proposed to relax the XT constraint. Then, to match the CC algorithm, we describe the crosstalk and fragmentation in the network and propose a comprehensive algorithm considering crosstalk and fragmentation (SCCF), which effectively improves the spectrum utilization rate. The proposed algorithm takes into account the influence of crosstalk problem and fragmentation problem on the network and the relationship between them. The simulation results show that the proposed CC-SCCF RMCSA algorithm reduces the XT of the lightpath and optimizes the blocking probability and spectral utilization.

The rest of this article is organized as follows. Work on the fragmentation problem and the XT problem is described in Section 2. Section 3 describes in detail the 3D XT fragmentation metric model. In Section 4, we propose the RMCSA algorithm based on CC-SCCF. We then evaluate the performance of the proposed algorithm in Section 5. Finally, the sixth part summarizes the full text.

## 2. Related Works

Firstly, some defragmentation algorithms have been proposed for the fragmentation problem in EONs. Thangaraj in [13] proposed an optimal spectrum defragmentation algorithm with a split spectrum approach. By effectively utilizing the minimum number of spectrum paths, the algorithm solved the limitation of multipath provisioning and reduced the extra spectrum consumption. Yadav et al. in [14] proposed a reactive defragmentation rerouting and spectrum assignment algorithm for when existing connection requests are terminated and network resources are released. Pan et al. in [15] proposed a multiple leaf-ringing-based protection algorithm with a spectrum defragmentation algorithm. When there are not enough available spectrum resources in the protection paths, the proposed algorithm triggers spectrum defragmentation and re-optimized resource utilization to improve efficiency in the network. However, these defragmentation algorithms may disrupt traffic, which adversely affects network operation and maintenance. Li et al. in [16] proposed a hitless deep reinforcement learning-based solution to achieve self-adaptive spectrum optimization. Selva et al. in [17] proposed a multiconstrained defragmentation algorithm to reconfigure the existing connections in a hitless manner. However, the hitless defragmentation algorithms struggle to build finer systems, meaning more complex optical components hardware costs are required.

The fragmentation-aware algorithms consider the impact of resource allocation on network fragmentation before allocating spectrum resources for requests, which effectively reduces fragmentation without disrupting traffic. Liu et al. in [18] proposed a fragmentation-aware routing and spectrum allocation based on a spectrum-slicing algorithm. A path weight formula and a spectrum slicing algorithm are proposed in the route selection stage and spectrum allocation stage, respectively to improve spectrum utilization. Bao et al. in [19] proposed a service-driven fragmentation-aware resource allocation scheme to enhance resource utilization by avoiding fragmentation with the joint consideration of the used path and neighboring links. Klinkowski et al. in [20] proposed a joint load-balanced and fragmentation-aware algorithm called LBFA to reduce fragmentation and improve spectrum utilization. Pourkarimi et al. in [21] proposed a core classification fragmentation-aware algorithm which could split resources according to the request demand and determine the appropriate spectral space using a cost function.

However, only considering the fragmentation will lead to a serious impact for the inter-core XT on the optical network, so the research on the inter-core XT is deepening. Similar to in the case of the fragmentation issue, in order to solve the XT issue, the XT-aware approach reduces the XT and prevents traffic disruption.

In addition, Ahmed et al. in [22] compared XT-aware and XT-avoided algorithms. The simulation result shows that, compared with the XT-avoided variety, XT-aware algorithms improve resource utilization better. Klinkowski et al. in [23] analyzed and compared schemes based on both the static/worst-case XT algorithms and the dynamic/exact XT estimation algorithms to ensure acceptable XT levels in the network. In [24], Zhang et al. proposed the Low-netwXT algorithm in SDM-EONs, which solved the RMCSA problem by the realization of crosstalk effect minimization and balance in MCF-based EONs.

Xiong et al. in [25] proposed a deep learning and hierarchical graph-assisted crosstalk-aware fragmentation avoidance algorithm. In this algorithm, a 3D measurement model is used to achieve more accurate results. Meanwhile, according to deep learning traffic predictions and the different characteristics of hierarchical graphs, an adaptive resource allocation algorithm is proposed, which considers eliminating core adjacency and reducing modulation format. However, machine learning algorithms cannot deal with emergencies well, and heuristic algorithms have certain advantages in this respect.

Lira et al. in [26] proposed a system design framework for the heuristic SA algorithm based on the MSCL principle and applied the proposed framework to design two new SA heuristics, namely MPAO-Wj and MPAO-WLj. Liu et al. in [27] proposed a routing spectral core assignment method (TMD-XT-RSCA) which takes into account hold time, lightpath neighborhood matching, and inner kernel crosstalk. The crosstalk issue is optimized by

balancing the load of the lightpath, balancing the load of the fiber core, and considering the time domain fragmentation. Chatterjee et al. in [28] introduced core, mode and spectrum (CMS) algorithm, which allocates resources in turn in terms of core, mode, and spectrum. However, the above studies on XT and fragmentation issues are not accurate enough and ignore the relation between XT and fragmentation needed to achieve better-integrated network performance.

Therefore, the main contribution of this paper is to optimize the existing 3D metric model in order to to accurately evaluate the current state of the network from the spatial domain, frequency domain, and time domain. In addition, the proposed CC-SCCF algorithm clarifies the relationship between XT and fragmentation in order to improve resource utilization and reduce the impact of XT in the network.

### 3. System Model and the Enabling Technologies

#### 3.1. Network Model

Generally, the network topology describes the MCF-based EONs, which need to start from three aspects: the network nodes, the links, and the fiber cores. Therefore, the network model is set to $G(V, E, C)$, which means there are $V$ nodes, and that there are $E$ links in the network, and that each link contains $C$ cores. In our model, each core can carry $|F|$ frequency slots, whose bandwidth is 12.5 GHz. Additionally, a connection request ($R^i$) is modeled as $R^i(s^i, d^i, b^i, t_s^i, t_e^i)$, which means the connection request from the source node $s^i$ to the destination node $d^i$ starts at $t_s^i$, ends at $t_e^i$ and requires $b^i$ (Gbps) bandwidth.

When $R^i$ reaches the network dynamically, the number of FSs $R^i$ needed is first calculated. The number of FSs is then used to determine whether $R^i$ is established or blocked by the lightpath ($P^i$) from node $s^i$ to $d^i$. The number of FSs required by $R^i$ is determined by the size of the bandwidth demand and the selected modulation level, which can be calculated by Equation (1) [29]

$$F^i = \left\lceil \frac{b^i}{C_{slot} * M} \right\rceil + GB, \tag{1}$$

where $F^i$ and $b^i$ represent the number of FSs and bandwidth required by $R^i$. $M$ denotes modulation level (i.e., 1, 2, 3, 4). $C_{slot}$ is the bandwidth granularity of each FS. $GB$ is the number of FS of the guard band required to ensure non-overlapping constraint. The FSs assigned to $R^i$ is expressed as $[f_s^i, f_e^i]$, where $f_s^i$ and $f_e^i$ are the indexes of start FS and end FS for $R_i$, respectively.

In addition to non-overlapping, when these resources are assigned to a lightpath, they should the satisfy constraints of spectrum continuity and contiguity. Indeed, switching between different cores is not allowed in the network nodes. That is, the selected core for the same lightpaths in all routing links is fixed to satisfy the constraint of spatial continuity constraints. As discussed in [29], these constraints must be followed during resource allocation in the MCF-based EONs.

#### 3.2. Three Domains XT-Fragmentation Metric Model

In this section, we describe a synthetic metric model in detail. To measure the network state more precisely, both the effects of XT and fragmentation on the network should be simultaneously described when describing the network with a model. Based on this, by further extending the existing 3D fragmentation metric model [30] a 3D XT-fragmentation metric model is proposed. We introduce the model from the spatial, frequency, and time domains.

#### 3.2.1. Spatial Domain

In this paper, the typical seven-core fiber with simple structures and clear characteristics in Figure 1a is selected [29]. In an MCF, in addition to the middle core, which is marked green, six other outer layer cores adjacent to the central core are marked in two colors, and the same color is not adjacent. Additionally, adjacent cores with different colors may be

affected by XT. This is because the closer the distance between the cores is, the more seriously affected the core is by the XT. Therefore, the XT between the non-adjacent cores is ignored.

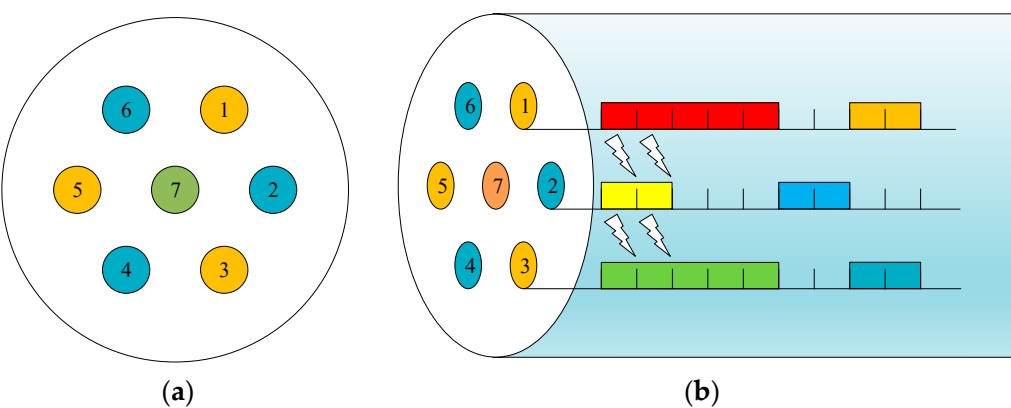

**Figure 1.** (**a**) a seven-core fiber and (**b**) effects of inter-core XT.

Specifically, when lightpath resources in each core of this link are occupied by requests, as shown in Figure 1b, the horizontal direction represents the FSs, the vertical direction represents cores, the colored parts represent the lightpath resources already occupied by requests, and the uncolored parts represent the idle resources. Since the same FS on core 2 and core 1 is occupied, these two cores are affected by the XT, which also affects core 3.

Therefore, for the XT issue in the spatial domain, Equation (2) is usually used to calculate the *XT* between the cores *i* and *j*:

$$XT = \frac{n - n\exp\left[-(n+1)\cdot 2\cdot h_{i,j}\cdot L\right]}{1 + n\exp\left[-(n+1)\cdot 2\cdot h_{i,j}\cdot L\right]}, \tag{2}$$

where $h_{i,j}$ and $L$ represent the power coupling coefficient and the fiber length, respectively. $h_{i,j}$ is related to multiple parameters in MCF, which is mathematically expressed as Equation (3):

$$h_{i,j} = \frac{2k_{i,j}^2 \cdot br}{\beta \cdot \Lambda_{i,j}}, \tag{3}$$

where $br$ and $\beta$ are the bending radius and the propagation constant, respectively, whereas $\Lambda_{i,j}$ and $k_{i,j}$ are the core pitch and the mode coupling coefficient between cores *i* and *j*, respectively. According to optical waveguide theory, $k_{i,j}$ is calculated by Equation (4):

$$k_{i,j} = \frac{\sqrt{\Delta}}{cr} \cdot \frac{U^2}{V^3} \cdot \frac{K_0\left(\frac{\Lambda_{i,j}}{cr}\cdot W\right)}{K_1^2(W)}, \tag{4}$$

where $\Delta$, $cr$, $W$, $V$, and $U$ represent the relative refractive index difference, the core radius, the cladding, the normalized frequency, and the normalized transverse wave numbers in the core, respectively.

For the fragmentation issue in the spatial domain, since the fragmentation between the cores does not affect the other cores, it is only necessary to consider the number of cores with fragmentation. In Figure 1b, when the request needs to occupy 3FSs, neither core 1 not 3 can allocate sufficient resources for the request, while core 2 can provide spectrum resources. However, if the request needs to occupy 4FSs, core 1,2,3 are affected by the fragmentation issue.

3.2.2. Frequency Domain

As mentioned above, when the adjacent different cores use the same FSs, they are affected by more serious XT. Therefore, in the spectrum domain, the overlap of the occupied resources in the adjacent cores needs to be considered and the results are obtained by summing up, as shown in Equation (5):

$$XT_{i,j}^F = \sum_{f\in F} XT_{i,j}\cdot y_{i,j}^f, \tag{5}$$

where $F$ is the set of FSs considered and $y_{ij}^{f}$ is a binary value: 1 when this FS on the core $i$ and $j$ is overlapped, and 0 when not overlapped.

Taking the time slot (TS) 1 of Figure 2 as an example, the resource distribution of core 1,2,3 is visible, and only the first 18 FSs are shown. With Equation (5), the inter-core XT between any two adjacent cores can be obtained. For example, for core 1 and 2, the XT $XT_{F\,1,2}$ between cores 1 and 2 is $XT_{F\,1,2} = 2 \cdot XT_{1,2}$ since the two lightpaths in these cores have two FSs in common. Similarly, the overall inter-core XT suffered by a specific core may be expressed as Equation (6):

$$XT_i^F = \sum_{f \in F, j \in C: j \neq i} XT_{i,j} \cdot y_{i,j}^{f}, \tag{6}$$

where C denotes all the cores in MCF. In Figure 2 TS1, the total inter-core XT of core 2 is $XT_2^F = 2 \cdot XT_{1,2} + 4 \cdot XT_{2,3}$. because for core 2, 2 FSs are affected by XT with core 1 and 2 FSs with core 3.

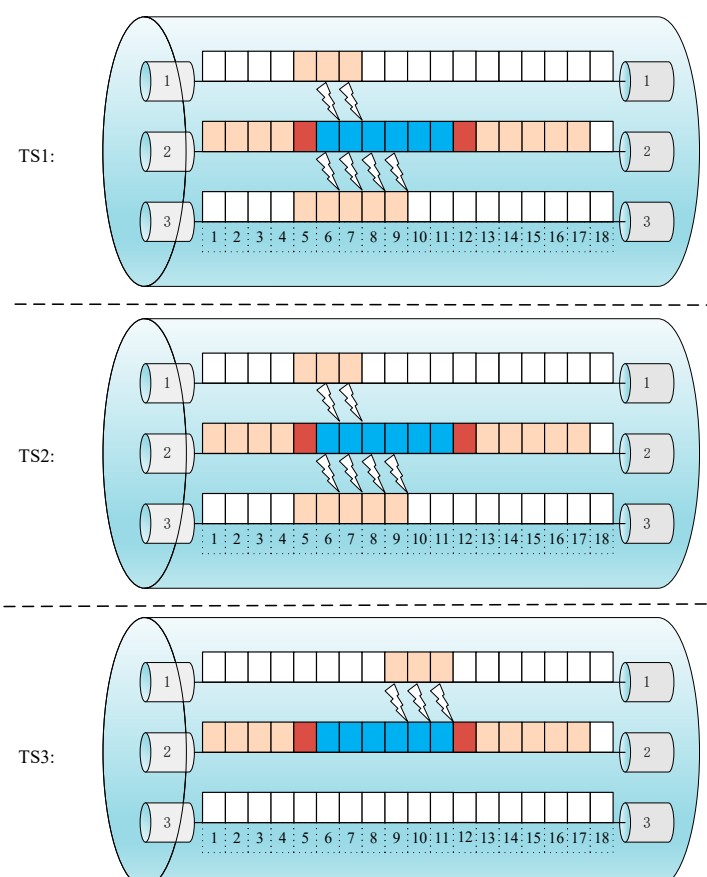

**Figure 2.** 3D inter-core XT model.

In terms of fragmentation, the intra-core fragmentation issue only affects the current core. The effect of the fragmentation $Fra_i^F$ on the frequency domain can also be obtained using a similar formula to Equation (6), as shown in Equation (7):

$$Fra_i^F = \sum_{f \in F} Fra_i \cdot y_i^{f}, \tag{7}$$

For core 2 in Figure 2 TS1, the red-colored FSs represent the idle but unavailable spectrum resources, which owe their condition to fragmentation. It can be seen that there are only two discrete free frequency gaps. Thus, for requests occupying FSs more than 1, the amount of fragmentation in the fiber core 2 is $Fra_{F\,2} = 2 \cdot F_2$.

### 3.2.3. Time Domain

When modeling $R^i$, both the $t_s^i$ and $t_e^i$ are considered, because the XT and fragmentation are only generated when the request occupies the resources in the network. Therefore, the time domain is also considered when computing the XT and fragmentation issues. In general, a longer lightpath request duration leads to more serious inter-core XT. Considering the time domain, we can further define the time-weighted inter-core XT as Equation (8):

$$XT_{i,j}^{T,F} = \sum_{t \in T} XT_{i,j}^{F} \cdot z_{i,j}^{t,f}, \tag{8}$$

where $T$ represents the set of TSs considered in the algorithm and $z_{i,j}^{t,f}$ is a binary value that denotes the lightpath occupation state of FS $f$ in cores $i$ and $j$ within TS $t$.

An example is shown in Figure 2, which denotes the FS usage in a 7-core fiber from TS1 to TS3. We compute the time-weighted XT using Equation (8). For example, the total XT between cores 1 and 2 is $XT_{F\,1,2} = 2 \cdot XT_{F\,1,2} + 2 \cdot XT_{F\,1,2} + 3 \cdot XT_{F\,1,2}$, because, for the core 2, 2 FSs are affected by XT in TS1, 2 FSs in TS2, and 3 FSs in TS3. At the same time, we can also calculate the XT effect of any core by using the three domains of spatial, frequency, and time with Equation (9):

$$XT_i^{T,F} = \sum_{t \in T, j \in C: j \neq i} XT_i^{F} \cdot z_{i,j}^{t,f}, \tag{9}$$

In Figure 2, the inter-core XT of core 2 is $XT_{T,F\,2} = (2 \cdot XT_{F\,1,2} + 4 \cdot XT_{F\,2,3}) + (2 \cdot XT_{F\,1,2} + 4 \cdot XT_{F\,2,3}) + (3 \cdot XT_{F\,1,2})$. As can be seen, the XT affects core 2 on the three TSs. In TS1 and TS2, two adjacent cores overlap requests on core 2, respectively: 2 FSs are affected in core 1 and 2 FSs are impactd in core 3. In TS3, 3 FSs are affected between cores 1 and 2.

Similarly, we can also obtain the amount of fragmentation within the fiber core by weighting using Equation (10):

$$Fra_i^{T,F} = \sum_{t \in T} Fra_i^{F} \cdot z_i^{t}, \tag{10}$$

By using Equation (10), the amount of fragmentation of core 2 can be easily obtained, $Fra_2^{T,F} (2 \cdot Fra_{F\,2}) + (2 \cdot Fra_{F\,2}) + (2 \cdot Fra_{F\,2})$

After defining the inter-core XT jointly in the spatial, frequency, and time domains, we can calculate the total XT and fragmentation for each established lightpath by using Equations (11)–(13).

$$XT_{i,j}^{F,T} = \sum_{f \in F} \sum_{t \in T} \frac{n - n \exp\left[-(n+1) \cdot 2 \cdot h_{i,j} \cdot L\right]}{1 + n \exp\left[-(n+1) \cdot 2 \cdot h_{i,j} \cdot L\right]} \cdot y_{i,j}^{f} \cdot z_{i,j}^{t}, \tag{11}$$

$$XT_i^{F,T} = \sum_{f \in F, j \in C, j \neq i} \sum_{t \in T, j \in C, j \neq i} \frac{n - n \exp\left[-(n+1) \cdot 2 \cdot h_{i,j} \cdot L\right]}{1 + n \exp\left[-(n+1) \cdot 2 \cdot h_{i,j} \cdot L\right]} \cdot y_{i,j}^{f} \cdot z_{i,j}^{t}, \tag{12}$$

$$Fra_i^{F,T} = \sum_{f \in F} \sum_{t \in T} Fra_i \cdot y_i^{f} \cdot z_i^{t}, \tag{13}$$

To ensure the lightpaths have better signal transmission qualities, we need to ensure that the total inter-core XT per FS per TS suffered on each of the links along the lightpath is no greater than a pre-defined threshold. If any link cannot guarantee this condition, then we should not allow this lightpath to be established on that path.

### 3.3. Routing and Modulation Format Selection

In routing, the choice of the modulation format is related to the transmission range, which in turn follows the XT threshold, as shown in Table 1 [24]. Therefore, we choose to use the classical k-shortest path (KSP) algorithm. Using a high-level modulation format can also reduce the consumption of spectrum resources and alleviate the fragmentation issue. However, the disadvantage of doing this is that the XT threshold is lowered, and our proposed CC algorithm can effectively alleviate this disadvantage. This paper considers four modulation formats, binary phase-shift keying (BPSK), quadrature phase-shift keying (QPSK), 8-quadrature amplitude modulation (QAM), and 16-QAM. Meanwhile, we con-

sider all the available modulation formats for each lightpath, as shown in Equation (14), where the highest available modulation format $M^i_{\max}$ is determined by the transmission reach. Then, a collaborative path for each modulation format is selected.

$$M^i = \left\{ M^i_1, M^i_2, \ldots, M^i_j, M^i_{\max} \right\}, \qquad (14)$$

where $i$ represents the K path obtained in the KSP algorithm and $j$ is the modulation format that can be adopted in each path.

**Table 1.** The parameters are used to determine the transmission reach, capacity per frequency slot, and XT thresholds for each modulation format.

| Modulation Formats | Transmission Reach [km] | Capacity pFS [GHz] | XT Thresholds [dB] |
|---|---|---|---|
| BPSK | 4000 | 12.5 | −14 |
| QPSK | 2000 | 25 | −18.5 |
| 8-QAM | 1000 | 37.5 | −21 |
| 16-QAM | 500 | 50 | −25 |

## 4. Crosstalk Classification Based on Synthetically Consider Crosstalk and Fragmentation Algorithm

In this section, we present a CC-SCCF RMCSA, which contains two components: the CC algorithm and the SCCF algorithm.

### 4.1. Crosstalk Classification Algorithm

In the traditional RSA algorithm, to improve the spectrum utilization radio and reduce the blocking radio, the requests occupy the spectrum resources as tightly as possible by different allocation strategies. However, this RSA algorithm will inevitably lead to the increased influence of XT in the network. When the traffic load is low, the spectrum resources in the network are relatively abundant, and the influence of the XT can be reduced by discrete resources in the network under the premise of satisfying the request transmission. Based on this, the CC algorithm is proposed to divide the XT threshold $XT$ into three stages based on the traffic load situation $0XT$. The second stage allows the production of partial $XT$, namely $\alpha XT$, where $\alpha$ is a weight which is used to test the extent to which relaxing the XT threshold affects the network. The third stage involves the transmission of the request while satisfying the XT threshold $XT$. When the volume of requests is small, the first stage should be adopted. As the volume increases, the available resources in the network decreases, leading to requests becoming blocked. When this occurs, XT stage upgrades to the second stage, until the upgrade to the third stage can be made.

Algorithm 1 shows the process of allocating resources for each lightpath request that reaches the network in the CC algorithm. When a new connection request $R^i(s^i, d^i, b^i, t^i_s, t^i_e)$ arrives, we first judge whether there is any lightpath to be released in the connection $R^{el}$. If so, we update $R^{el}$ and release the occupied spectrum resources (steps 1–5). Then, according to the KSP algorithm, the k-shortest paths are selected for $R^i$ and stored in the $P^i$ (steps 6). Next, the available modulation level ($M^i_j$) is selected for each path in $P^i$, where the highest modulation level ($M^i_{max}$) is determined by transmission distance. For each $M^i_j$, the number of $R^i$-occupied FSs ($F^i$) is calculated according to Equation (1) (steps 7–12). By combining $P^i$ and $M^i_j$, the collaboration path ($CP^i$) is obtained and is numbered in ascending order of $F^i$ (step 13). After that, we set the XT stage in $CP^i$ as $CS^i$ (steps 14–15). Next, we determine whether the core (c) in $CP^i$ contains appropriate idle spectrum blocks (SBs) under the $CS^i$ and store the SBs as $B^i$ (steps 16–18). Then, Algorithm 2 is used to determine whether $B^i$ meets the crosstalk constraint. If $B^i$ meets the crosstalk constraint, $B^i$ is stored as an alternative spectrum block ($AB^i$) (steps 19–28), and $AB^i$ with the minimum degree ($d^i$) of XT effect is allocated for $R^i$ (steps 29–30). Otherwise, $R^i$ will be blocked after the total paths are searched (steps 31–34).

| **Algorithm 1**: CC algorithm |
|---|

**Input**: Arriving connection request $R^i(s^i, d^i, b^i, t_s^i, t_e^i)$.
**Output**: Spectrum allocation.
1:      **for** each existing connection $R^{el}(s^{el}, d^{el}, b^{el}, t_s^{el}, t_e^{el})$ do
2:            **if** $t_s^{el} < t_e^{el}$ then
3:                  Update $R^{el}$ and release the occupied spectrum resources.
4:            **end if**
5:      **end for**
6:      Select the *k*-shortest paths for $R^i$ according to the KSP algorithm and store in the $P^i$
7:      **for** each $P^i$ do
8:            Determine the highest modulation format level $M_{max}^i$.
9:                  **for** each modulation level $M_j^i$ in $[M_1^i, M_{max}^i]$ do
10:                        Compute the number of FSs $F_i$ requested by Equation (1).
11:                  **end for**
12:            **end for**
13:      $CP^i$ is numbered in ascending order of $F^i$
14:      **for** each collaboration path $CP^i$ do
15:            Set the XT stage as $CS^i$.
16:            **for** each $CS^i$ do
17:                  **for** each c of $CP^i$ do
18:                        According to the size of SBs and the crosstalk threshold, search
                          the available SBs as $B^i$.
19:                        **if** $B^i \neq$ None then
20:                              **for** each $B^i$ do
21:                              Compute XT based on Algorithm 2.
22:                                    **if** Algorithm 2 returns 1 then
23:                                          Store the $B^i$ in $AB^i$
24:                                          **Break**.
25:                                    **end if**
26:                              **end for**
27:                        **end if**
28:                  **end for**
29:                  **if** $AB^i \neq$ None then
30:                        Select the SB with the first minimum of $d^i$ and allocate the $d^i$ for $R^i$.
31:                  **end if**
32:                  Reject connection request $R^i$
33:            **end for**
34:      **end for**

### 4.2. Synthetically Consider Crosstalk and Fragmentation Algorithm

Since the CC algorithm displays better XT optimization performance at a low load, the growth of XT cannot be suppressed while in the third stage. To fit with the CC algorithm better, the SCCF algorithm measures the 3D model and simultaneously measures the effects of XT and fragmentation on the network. First, we simplify Equation (11) so that both XT and fragmentation are represented by frequency gap numbers

$$XT_{i,j}^{F,T} = \sum_{f \in F} \sum_{t \in T} XT_{i,j} \cdot y_{i,j}^f \cdot z_{i,j}^t, \tag{15}$$

where $XT_{i,j}$ denotes the number of FSs in which affected by XT.

Then, combined with Equation (13), the current combined impact ($CI_i^{F,T}$) of the core on the network is:

$$\begin{aligned} CI_i^{F,T} &= XT_{i,j}^{F,T} + Fra_i^{F,T} \\ &= \sum_{f \in F} \sum_{t \in T} XT_{i,j} \cdot y_{i,j}^f \cdot z_{i,j}^t + \sum_{f \in F} \sum_{t \in T} Fra_i \cdot y_i^f \cdot z_i^t, \end{aligned} \tag{16}$$

Based on Equations (15) and (16), the effects of the core and the adjacent core can be found, respectively. During resource allocation, the most appropriate spectrum resources can be allocated to requests by measuring the network state multiple times, as in Algorithm 2.

| **Algorithm 2**: SCCF algorithm |
|---|

| **Input**: $P^i$, $c^i$, $f_s^i$, $f_e^i$. |
|---|
| **Output:** 1 or 0. |
| 1:     $XT = 0$. |
| 2:     **for** FS $f^i$ in $[f_s^i, f_e^i]$ do |
| 3:         **for** (link $e^i$, core $c^i$) in $P^i$ do |
| 4:             **for** each adjacent $c_a^i$ of $c^i$ do |
| 5:                 **if** $R^{el} \neq$ None then |
| 6:                     **for** each $e^{el}$ do |
| 7:                         Calculate $XT_{i,j}^{F,T}$, $XT_i^{F,T}$, and $CI_i^{F,T}$. |
| 8:                     **end for** |
| 9:                     **if** $XT_i^{F,T} > XT_{threshold}$ then |
| 10:                         **return** 0 |
| 11:                     **end if** |
| 12:                 **end if** |
| 13:             **end for** |
| 14:         **end for** |
| 15:         Select the spectrum resource with minimal $CI_i^{F,T}$ |
| 16:         **if** $XT_{i,j}^{F,T} > XT_{threshold}$ then |
| 17:             **return** 0 |
| 18:         **end if** |
| 19:     **end for** |
| 20:     **return** 1 |

To calculate the size of crosstalk and fragmentation, the inputs to Algorithm 2 are the network state on the path, including the pre-assigned routing path $P^i$ on the path, core $c^i$, the start FS $f_s^i$, and the end FS $f_e^i$. The output of the algorithm is a Boolean number: 1 indicates that the new lightpath can be successfully transmitted over these pre-allocated FS, and 0 indicates that the lightpath cannot be transmitted.

As a new Pi arrives, we determine the location of the SBs according to $[f_s^i, f_e^i]$, $P^i$, and $c^i$ (steps 1–4). Then, we determine the occupation status of adjacent cores at the $f^i$ of $R^{el}$ transmission (steps 5–7). If the spectrum resources are being occupied by lightpaths, we calculate the XT $XT_{i,j}^{F,T}$ by Equation (16) and check if it meets the $XT_{threshold}$ to ensure transmission successfully (steps 8–14). Finally, we compute and judge whether the XT on each FS assigned for $P^i$ is less than the $XT_{threshold}$, and select the SB with the lowest total crosstalk effect (steps 15–20).

Finally, the XT on each assigned FS of $P_i$ itself is calculated and checked to determine if it is less than $XT_{threshold}$ and choose the spectrum resources with minimal combined impact (steps 15–20).

### 4.3. Complexity

The time complexity of the algorithm is determined by the number of loops in the algorithm. The time complexity in the CC-SCCF algorithm derives from the CC algorithm and the SCCF algorithm. The CC algorithm traverses spectrum resources in different crosstalk stages of all collaboration paths. The time complexity of planning collaboration paths for each request is $O(k|M||E|\log|V|)$, the complexity of the XT stages is $O(|N|)$, the complexity of traversing the spectrums is $O(|C||S|)$. Among them, k indicates the number of the shortest paths, whereas $|M|$, $|V|$, $|E|$, $|N|$, $|C|$ and $|S|$, respectively, indicate the number of modulation levels being considered, nodes, links, XT stages being set, cores in an MCF link and FSs in each core in the network. The SCCF algorithm calculates and determines the crosstalk of each FSs, whose time complexity can be rendered as $O(|E||C||S|)$. As a result, the CC-SCCF algorithm's time complexity is $O(k|C|^2|E|^2|M||N||S|^2\log|V|)$.

## 5. Simulation Results and Analysis

In this section, we evaluated the performance of the CC-SCCF RMCSA algorithm in two different network environments: the COST239 network with 11 nodes and 26 links [29] and the NSFNET network with 14 nodes and 21 links [30,31], as shown in Figure 3. The

colored circles in Figure 3 represent nodes, and the numbers beside the links represent the distance [km] between nodes. Additionally, each core in the links contains 358 FSs of 12.5 GHz [32,33]. As shown in Figure 1, each link is an MCF consisting of 7 cores. The bandwidth of each lightpath follows an even distribution between 50 GHz and 400 GHz [34], and dynamically arriving optical path requests are subject to a negative exponential distribution, with an average of 100 ms [35]. The parameters of transmission reach, capacity pFS and inter-core XT threshold for the four common modulation formats are shown in Table 1 [24]. The protection bandwidth is set to two FSs to prevent interference effects between adjacent lightpaths [36,37]. Since only one kind of physical layer impact, inter-core XT, is considered in the algorithm, we assume that, when the XT threshold is met, services can be transmitted successfully. The CC-SCCF algorithm is compared with the algorithm using only the CC algorithm and the SCCF algorithm, which highlights the fit of the CC-SCCF algorithm, which is compared with the CMS RMCSA algorithm to reflect the algorithm performance [38]. The CMS algorithm is a traditional algorithm in SDM-EONs which allocates resources sequentially from core, mode and spectrum. Considering that MCF does not require mode allocation, we simplify the CMS algorithm into a resource allocation algorithm that only considers the core and spectrum. We also use the results of the first hits as a contrast, being those of a standard performance.

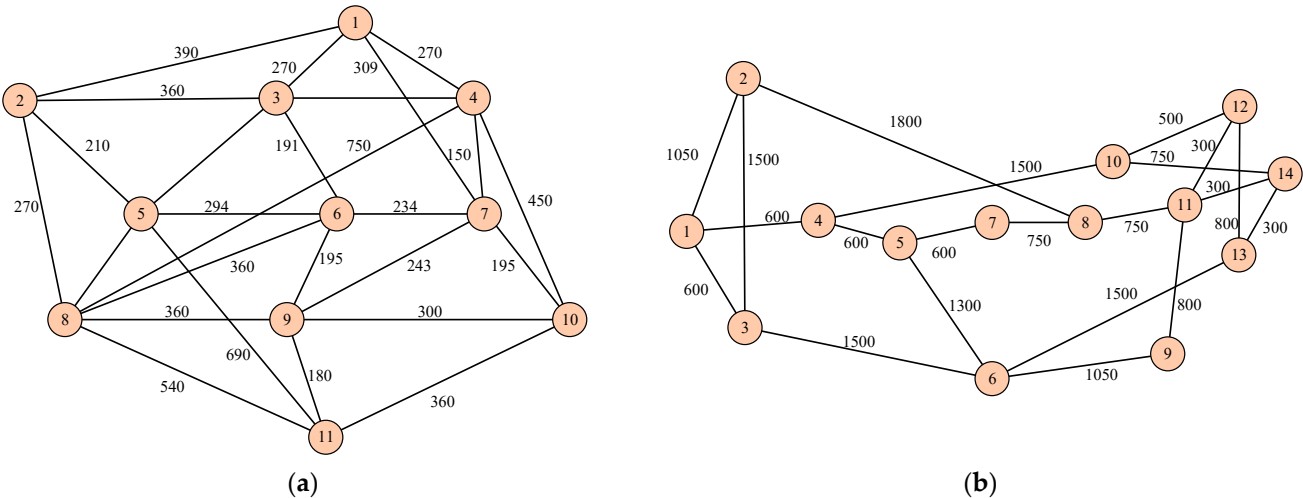

**Figure 3.** Test networks of (**a**) 11-node, 26-link COST239 and (**b**) 14-node, 21-link NSFNET.

We introduce the weight $\alpha$ when introducing the CC algorithm in 4.1 to determine the XT threshold in the second stage, which directly affects the performance of the CC-SCCF algorithm. Since the RSA problem is an NP-hard problem, the RMCSA problem extended by RSA is also an NP-hard problem, and so it is difficult to obtain accurate weights by using mathematical theoretical model. Therefore, we obtain a relatively optimal weight through simulation. Therefore, we first determine the changes of network performance caused by the value of $\alpha$ in different network environments, through simulation, as shown in Figure 4. It can be observed that the combined impact ratio has the worst performance when $\alpha$ values of 0 and 1 are both present in NSFNET and COST239 networks. When $\alpha = 0$ and 1, the second phase of the CC algorithm fails. This in turn affects the SCCF algorithm, so the overall optimization of the network is poor.

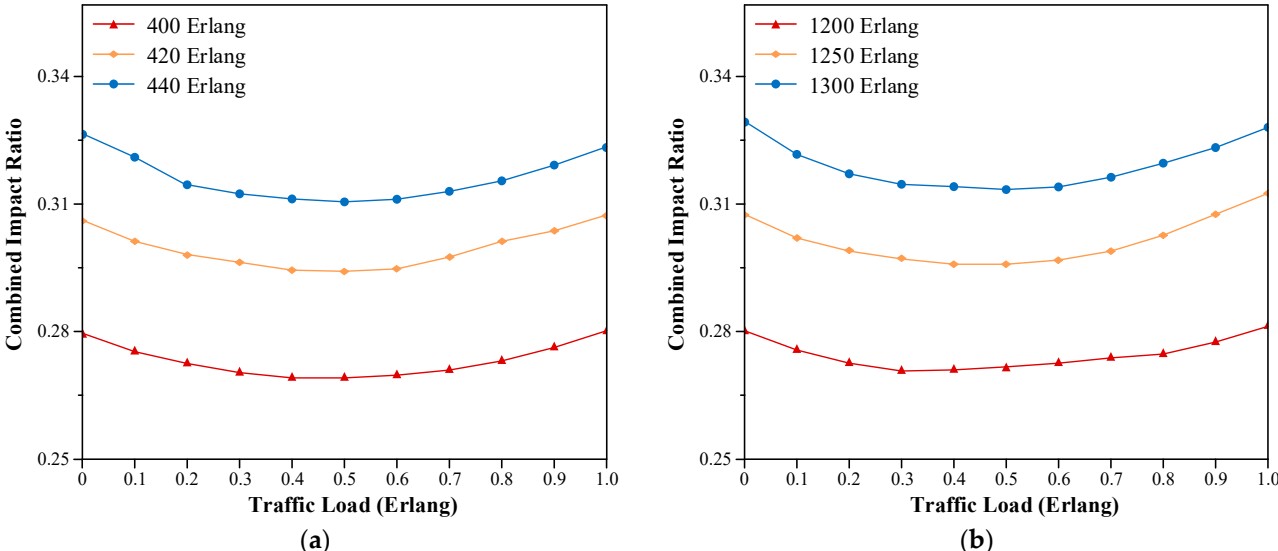

**Figure 4.** Comparison of Combined Impact Ratio for different weights ($\alpha$) in (**a**) COST239 and (**b**) NSFNET.

Meanwhile, we observed that the lowest combined impact ratio can be obtained by adopting different $\alpha$ values under different traffic loads. Specifically, as the network traffic load increases, the blocking probability is optimal. It is suitable for NSFNET and COST239, i.e., for NSFNET 400, 420 and 440 [Erlang], 0.4, 0.5 and 0.5 are the best. In the COST239 network, 0.3, 0.4, and 0.5 are the best for 1200, 1250 and 1300 [Erlang]. This is because the number of incoming requests increases per unit of time as the traffic load increases. To ensure the fairness of the performance comparison between the CC-SCCF algorithm and the benchmark, a fixed weight value ($\alpha = 0.5$ in the NSFNET network and $\alpha = 0.4$ in the COST239 network) is selected for all the SDFA algorithms.

## 5.1. Performance Comparison of Blocking Probability

In this subsection, we first compare the blocking probability (BP) of five algorithms. BP represents the ratio of the number of blocked connection requests to the total number of connection requests that arrive in the network. As shown in Figure 5a,b, the BP increases with increasing traffic load in the two networks. This is because the number of incoming connection requests per unit time increases as the traffic load increases. By comparing the CC, SCCF and CC-SCCF algorithms, it can be found that the CC-SCCF algorithm, combined with the CC algorithm and SCCF algorithm, shows better performance. In comparison with the first-fit (FF) algorithm, with the increase in traffic load, the performance of the CC algorithm is significantly improved. This is because, when the traffic load is small, the CC algorithm adopts $0XT$, which leads to a more serious fragmentation issue. Conversely, in $\alpha XT$ and $XT$ algorithms, the fragments are reduced and the blocking rate is reduced.

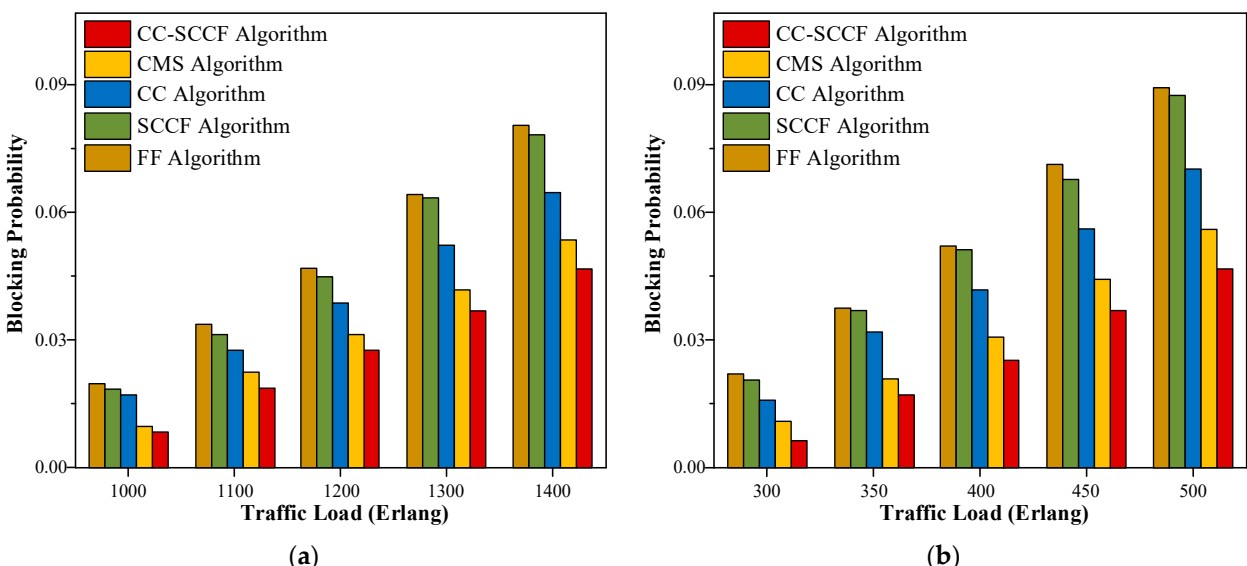

**Figure 5.** Comparison of blocking probability in (**a**) COST239 and (**b**) NSFNET.

In comparison with the CMS and CC-SCCF algorithms, it can be seen that the performance of the CC-SCCF algorithm improves by about 12% to 14% in high-traffic loads. This is because the CC-SCCF algorithm using the 3D metric model can describe the network state more accurately and utilize the idle spectrum more efficiently. At the same time, the algorithm calculates $Fra_i^{T,F}$, senses the impact of fragmentation on the network in advance, and reduces BP.

### 5.2. Network-Wide XT Effect Ratio

Due to the dynamic establishment and release of lightpaths, the XT of requests in the network are constantly changed. Therefore, compared with calculating the impact of a single request XT, calculating the network-wide XT can more clearly and stably reflect the influence of crosstalk.

We define the network-wide XT effect ratio as the ratio of the current XT effect degree to the maximum occupied degree of all spectrum resources. Figure 6a,b compare the network-wide XT effect ratio using five algorithms in COST239 and NSFNET networks, respectively. Compared with other algorithms, the CC-SCCF algorithm has the lowest XT effect, which is mainly caused by the CC algorithm. The comparison with the FF algorithm shows that the XT effect of the CC algorithm decreases with the increase in traffic load. After the connection request arrives, the 0XT stage is judged preferentially to minimize the XT effect generated by each request. Therefore, the XT effect in the network is limited to a relatively low state.

In comparison with the CMS algorithms, the CC-SCCF algorithm optimizes the XT rate by 8% and 11%, respectively. This is because the CC-SCCF algorithm reduces XT in the network at the two stages of $0XT$ and $\alpha XT$. Conversely, in the $XT$ stage, because the XT effect is calculated in advance by Equation (15), there is no serious XT in the distribution of requests.

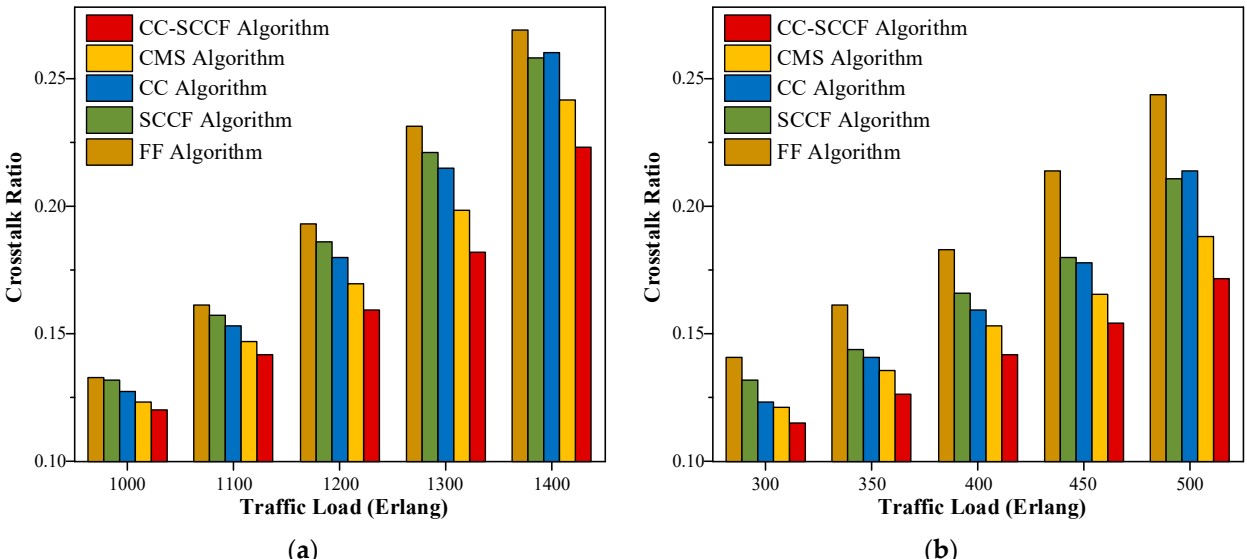

**Figure 6.** Comparison of XT effect in (**a**) COST239 and (**b**) NSFNET.

## 6. Conclusions

In this paper, a CC-SCCF resource allocation algorithm is proposed, which satisfies the XT constraint and improves the overall performance of the network. To achieve this goal, a 3D XT-fragmentation metric model is used to describe the more accurate current network state. Then, the CC algorithm is proposed to adjust the XT constraint on the request according to the current network state. By adjusting the XT constraint when the load is low, XT can be effectively reduced. When the load is high, the QoT of requests can be satisfied. Meanwhile, SCCF RMSA is proposed based on the CC algorithm, and the influence of XT and fragmentation on the network is considered. The effectiveness of the proposed algorithm is verified by the performance evaluation in COST239 and NSFNET networks. The simulation results show that the CC-SCCF RMCSA algorithm can effectively reduce the blocking probability and improve the spectrum utilization rate under the two network topologies.

**Author Contributions:** Conceptualization, Y.C. and J.Z.; methodology, Y.C.; software, Y.C.; validation, Y.C.; formal analysis, Y.C. and N.F.; investigation, Y.C. and Y.Z.; resources, Y.C.; data curation, Y.C.; writing—original draft preparation, Y.C. and N.F.; writing—review and editing, Y.C. and N.F.; visualization, Y.C. and Y.Z.; supervision, N.F.; project administration, D.R.; funding acquisition, J.Z. All authors have read and agreed to the published version of the manuscript.

**Funding:** This research received no external funding.

**Institutional Review Board Statement:** Not applicable.

**Informed Consent Statement:** Not applicable.

**Data Availability Statement:** The data in this paper is not publicly available at this time.

**Conflicts of Interest:** The authors declare no conflict of interest.

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
