# Peer review of "Crosstalk Classification Based on Synthetically Consider Crosstalk and Fragmentation RMCSA in Multi-Core Fiber-Based EONs"

_photonics, doi:10.3390/photonics10030340_

Round 1

Reviewer 1 Report

1. In this paper, the proposed algorithm is first to determine the XT constraint and then to maximize the spectrum utilization. How to consider the step order in the algorithm? If the two steps are reversed, is it applicable? For example, determine the block probability threshold first and then optimize the crosstalk ratio.

2. How do the modulation formats in Table I influence the results in Figs. 5 and 6? Are there any parameter values in Table I employed in the simulations?

3. The accuracy of traffic load in Fig. 4 is concerned. For example, is that possible that the minimum ratio happens at a value between 0.4 and 0.5 for erlang = 400 in Fig. 4(b)?

4. The erlang values selected for the two network topologies in Figs 4-6 are different. Why not select identical values for fair comparison?

5. Many grammatical/syntax issues should be correct:

Some phrases in the middle of a paragraph need to be correctly capitalized, for example, seeing lines 14 and 21.

MCF is first shown in line 38, but it is without defined.

Some abbreviations are not correctly expressed, for example, RMSA in line 42 and QoT in line 56.

The CMS algorithm used in the simulations is not defined.

The reference numbers in the main text do not match the ones on the list. 

Author Response

JIJUN ZHAO Research group

School of Information and Electrical Engineering

Hebei University of Engineering

Email: zjijun@hebeu.edu.cn

8 March 2023

Dear Reviewer 1

Photonics

Thank you for your valuable suggestion and the Reviewer's comments concerning our manuscript entitled “Crosstalk Classification based on Synthetically Consider Crosstalk and Fragmentation RMCSA in Multi-core Fiber-based EONs” (Manuscript Number: photonics-2251162). The paper has been improved a lot. We have revised the manuscript according to the recommendations and have provided detailed responses to each comment below. For the reviewers’ convenience, we have highlighted major changes in the revised manuscript in yellow background.

Reply to Reviewer's comments:

Reviewer 1:

  1. In this paper, the proposed algorithm is first to determine the XT constraint and then to maximize the spectrum utilization. How to consider the step order in the algorithm? If the two steps are reversed, is it applicable? For example, determine the block probability threshold first and then optimize the crosstalk ratio.

Re: Thank you very much for your valuable suggestion.

For the order of crosstalk constraint and spectrum utilization, the request first needs to meet the crosstalk threshold to ensure the quality of transmission. On the basis of meeting the crosstalk threshold, it is meaningful to reduce the fragmentation in the network, improve the resource utilization rate, and increase the quantity of transmissions. In fact, after satisfying the crosstalk constraint, we further comprehensively consider the impact of fragmentation issue and crosstalk issue through Eq. (16).

If the order is reversed, the combined crosstalk and fragmentation are considered first and the crosstalk threshold is considered later, the spectrum resources with less combined influence may not be able to transmit request because it does not meet the crosstalk threshold.

We add the following explanation to the 1. Introduction of the manuscript:

Lightpaths can be established and transmitted on the network only when requests meet the crosstalk constraint. Therefore, our algorithm will first consider meeting the crosstalk threshold to ensure the quality of transmission, and then further consider the comprehensive impact of crosstalk issue and fragmentation issue on the request.

  1. How do the modulation formats in Table I influence the results in Figs. 5 and 6? Are there any parameter values in Table I employed in the simulations?

Re: Thank you very much for your valuable suggestion.

(1) Both blocking probability and corsstalk ratio are related to the number of frequency slots required by services. Eq. (1) shows the relationship between the size of the frequency slots required and the modulation format. When the bandwidth required by the service is fixed, the higher the level of the modulation format used, the less the number of frequency slots required. If there are certain resources in the network, less resources required by connection request means the network can accommodate more services, and resource allocation causes lower blocking probability. It also means less crosstalk with other businesses, resulting in lower crosstalk ratio.

(2) The modulation format is related to the transmission distance, service FSs number, and crosstalk threshold. When the distance between nodes in the topology is long, the higher-order modulation format cannot be used. As a result, the number of FSs required by services increases. This not only affects the spectrum utilization and fragmentation problems, but also has different erlang values in the simulation results of different topologies when the FSs number is the same. Different modulation formats will also lead to different crosstalk threshold, which will also have certain influence on blocking rate. To verify our conclusion, the proposed algorithm is adjusted to only consider BPSK in NSFNET, and the simulation results are shown in Fig. 1. Both blocking probability and corsstalk ratio of the proposed algorithm increase obviously without the flexibility of modulation format.

(a) (b)

Figure 1. Comparison of Combined Impact Ratio for different weights (α) in (a) COST239 and(b) NSFNET

(3) We add the following contents to the manuscript:

In the pseudo-code of algorithm 1, We modify the line 18 as follows:

According to the size of SBs and the crosstalk threshold, search the available SBs as Bi.

  1. The accuracy of traffic load in Fig. 4 is concerned. For example, is that possible that the minimum ratio happens at a value between 0.4 and 0.5 for erlang = 400 in Fig. 4(b)?

Re: Thank you very much for your valuable suggestion.

(1) The minimum value in Fig.4 (b) is likely to appear between 0.4 and 0.5. Since the RSA problem is an NP-hard problem, the RMCSA problem extended by RSA is also an NP-hard problem, so it is difficult to use mathematical theoretical model to get accurate weights. We use simulation and get a relatively optimal weight, but this is not accurate. Therefore, we re-simulated experiments 0.42, 0.44, 0.46 and 0.48 within the range of 0.4 to 0.5, in which 420 Erlang is used in COST239 network and 1250 Erlang is used in NSFNET network. The test results show that the graph changes gently between 0.4 and 0.5, which has little impact on the following simulation results, as shown in Fig. 2.

(a) (b)

Figure 2. Comparison of Combined Impact Ratio for different weights (α) in (a) COST239 and(b) NSFNET

(2) We add the following contents to the 5. Simulation results and analysis in the manuscript:

Since the RSA problem is an NP-hard problem, the RMCSA problem extended by RSA is also an NP-hard problem, so it is difficult to get accurate weights by using mathematical theoretical model. Therefore, we get a relatively optimal weight through simulation.

  1. The erlang values selected for the two network topologies in Figs 4-6 are different. Why not select identical values for fair comparison?

Re: Thank you very much for your valuable suggestion.

In terms of distance between nodes, NSFNET network is much longer than COST239 network, which means it is difficult for NSFNET network to select high modulation formats, such as 16-QAM and 8-QAM. According to Table 1, the capacity of each FSs varies depending on the modulation format. Services in NSFNET network that cannot adopt highr modulation format will occupy more spectrum resources. When the network capacity is set to the same number of FSs (in our simulation, each link contains 358 FSs), the number of services that can be carried by NSFNET network decreases. Blocking probability is usually less than 0.1 in simulation work, so the two topologies select different erlang values.

  1. Many grammatical/syntax issues should be correct:

Re: Thank you so much for your careful check. We have scrutinized the revised manuscript and modified all errors which have been found. All of the revised sentences are highlighted in the revised manuscript.

Some phrases in the middle of a paragraph need to be correctly capitalized, for example, seeing lines 14 and 21.

Re: Thank you so much for your careful check. We changed Inter-core to inter-core in the 14 line and Crosstalk to crosstalk in the 21 line.

MCF is first shown in line 38, but it is without defined.

Re: Thank you so much for your careful check. We rewrite this passage on line 38 as:

Space division multiplexing (SDM) based on multi-core fiber (MCF) can effectively expand optical fiber capacity by utilizing spatial dimension [4]. Thus, SDM-EONs, which combine MCF-based SDM and EONs, have become one of the effective solutions to overcome the capacity crisis [5].

Some abbreviations are not correctly expressed, for example, RMSA in line 42 and QoT in line 56.

Re: Thank you so much for your careful check.

We modify routing, modulation format, and spectrum allocation to routing, modulation, and spectrum allocation.

We modify transmission quality to quality of transmission.

The CMS algorithm used in the simulations is not defined.

Re: Thank you so much for your careful check. We add a more detailed introduction in 5. Simulation results and analysis.

CMS algorithm is a traditional algorithm in SDM-EONs, which allocates resources sequentially from core, mode and spectrum. Considering that MCF does not require mode allocation, we simplify the CMS algorithm into a resource allocation algorithm that only considers the core and spectrum.

The reference numbers in the main text do not match the ones on the list.

Re: Thank you so much for your careful check. We mention that the error in reference 17 is found by checking and have been modified.

[17] Selva Kumar S, Kamalakannan J, Seetha R, et al. The Effectual Spectrum Defragmentation Algorithm with Holding Time Sensitivity in Elastic Optical Network (EON). International Journal of Optics, 2022, 2022.

Reviewer 2 Report

This article proposes two RMCSA algorithms that jointly decrease blocking and crosstalk in MCF-EONs. The crosstalk is measured in three dimensions (frequency, spatial, and time), and the threshold for crosstalk is adjusted with the variation of the load. The first algorithm assigns slots to the area that leads to the minimum crosstalk if the crosstalk is under the XT threshold. The second algorithm considers spectral fragmentation and attempts to jointly mitigate fragmentation and crosstalk. The paper is well-written with clear structure.

The paper introduced the constraints in MCF-EONs and the 3D crosstalk model. The effect of crosstalk associated with the current load in the network is considered.  The algorithm reduced blocking probability and crosstalk ratio jointly.

It will be better if the paper can improve with the following comments:

1. Failure to elaborate on why crosstalk can be divided into three stages (0XT, αXT, XT) when the loads are different.

2. The novelty is not very impressive. You may add extra work or comment the possible further development.

Author Response

JIJUN ZHAO Research group

School of Information and Electrical Engineering

Hebei University of Engineering

Email: zjijun@hebeu.edu.cn

8 March 2023

Dear Reviewer 2

Photonics

Thank you for your valuable suggestion and the Reviewer's comments concerning our manuscript entitled “Crosstalk Classification based on Synthetically Consider Crosstalk and Fragmentation RMCSA in Multi-core Fiber-based EONs” (Manuscript Number: photonics-2251162). The paper has been improved a lot. We have revised the manuscript according to the recommendations and have provided detailed responses to each comment below. For the reviewers’ convenience, we have highlighted major changes in the revised manuscript in yellow background.

Reply to Reviewer's comments:

Reviewer 2:

This article proposes two RMCSA algorithms that jointly decrease blocking and crosstalk in MCF-EONs. The crosstalk is measured in three dimensions (frequency, spatial, and time), and the threshold for crosstalk is adjusted with the variation of the load. The first algorithm assigns slots to the area that leads to the minimum crosstalk if the crosstalk is under the XT threshold. The second algorithm considers spectral fragmentation and attempts to jointly mitigate fragmentation and crosstalk. The paper is well-written with clear structure.

The paper introduced the constraints in MCF-EONs and the 3D crosstalk model. The effect of crosstalk associated with the current load in the network is considered.  The algorithm reduced blocking probability and crosstalk ratio jointly.

It will be better if the paper can improve with the following comments:

  1. Failure to elaborate on why crosstalk can be divided into three stages (0XT, αXT, XT) when the loads are different.

Re: Thank you very much for your valuable suggestion.

To be clear, the proposed crosstalk stage strategy is independent of traffic load. When the number of request is small, the spectrum resources in the network are relatively sufficient. In this case, even if the frequency spectrum utilization is reduced, the service blocking will not be caused and the impact of crosstalk can be reduced. We set the crosstalk for the low-traffic network which allow resource wastage to 0XT. However, when there is a large amount of request in the network, spectrum resources in the network are insufficient, which may cause service blocking. Therefore, in order to maximize the use of spectrum resources, we set the service just meet the crosstalk constraint. When there are a lot of traffic on the network, the crosstalk for the service is the crosstalk threshold, namely XT. In the interval of 0XT and XT, due to a certain degree of conflict between the crosstalk problem and the fragmentation problem, we propose a weight α for adjusting the crosstalk constraint, and try to find a weight that can balance the inter-core crosstalk and spectrum fragmentation through simulation. Therefore, crosstalk is divided into three stages.

  1. The novelty is not very impressive. You may add extra work or comment the possible further development.

Re: Thank you very much for your valuable suggestion. We deeply analyze and summarize the innovative contributions of the proposed algorithms, and add them to the introduction part of the revised manuscript.

The innovative contributions are as follows:

(1) The relationship between crosstalk problem and fragmentation problem is usually not considered in the existing researches when considering the optical network of space division multiplexing. As a result, only one side is optimized while the deterioration of the other side is ignored in the optimization of network performance. The proposed algorithm takes into account the influence of crosstalk problem and fragmentation problem on the network and the relationship between them, and gets the effect of overall network performance optimization.

(2) Extend the existing three-dimensional crosstalk model, and obtain the three domains crosstalk-fragmentation metric model which accurately describes the dynamic state of the network from three domains of spatial, frequency and time domain, realizing the simultaneous evaluation of the influence of inter-core crosstalk and spectrum fragmentation in the network, laying a foundation for the following work.

(3) Propose the CC algorithm. By adjusting the crosstalk threshold, the resource allocation strategy is dynamically adjusted according to the state of the network. Propose the SCCF algorithm. By measuring the influence of crosstalk problem and fragmentation problem comprehensively, the effectiveness of the algorithm is verified by the performance evaluation on COST239 and NSFNET networks. Blocking probability of the CC-SCCF algorithm is reduced about 13% and crosstalk ratio is reduced about 9% compared with CMS algorithm.

Reviewer 3 Report

The authors depart from the fact that in multi-core elastic optical networks, minimizing inter-core crosstalk implies not to use the same spectral band as a communication channel in one of two, spatially close neighboring cores; hence, the idle channel; i.e., the non-used spectral band, remains as a fragment until the neighbor core is not using it. In this sense, not all the spectral bands or wavelength-channels can be used at the same time in all cores. This is particularly true between cores that are spatially close.

The authors developed an algorithm, for the first time as they claim, that administrates the spectral-band assignation for a certain communication channel in such a way that the cross-talk is avoided (as much as possible) and the spectral fragmentation minimized; as a consequence, most of the spectral channels might be used at the same time in a multicore communication system.

Their report is well organized, the state of the art has been sufficiently revised and exposed, the concepts are clear; the hypotheses, formulas, programming and simulations make sense; the results compared to other works are clearly established and their simulations in quasi real environments are quite convincing.

In my opinion, the work is worth to be published in Photonics. 

My general comments on the contents of the manuscript are:

Line 38: Please re-write “SDM-EONs based on MCF combined with space division multiplexing (SDM) technology of multi-core fiber (MCF)…crisis [4,5].”

Line 107. “Selva et al. in [17]” is not valid as such paper in not in references.

Line 234: donates or denotes? {please revise}

Figure 2.3: What is the meaning of the red colored Frequency slots? Please explain within the text as they are somehow immune to the considerations.

Lines 298-299: please clarify the word radio (ratio?)

Lines 356-357: please re-write the phrase “The time complexity in the CC-SCCF algorithm is consisted of the CC algorithm and the SCCF algorithm.”

Author Response

JIJUN ZHAO Research group

School of Information and Electrical Engineering

Hebei University of Engineering

Email: zjijun@hebeu.edu.cn

8 March 2023

Dear Reviewer 3

Photonics

Thank you for your valuable suggestion and the Reviewer's comments concerning our manuscript entitled “Crosstalk Classification based on Synthetically Consider Crosstalk and Fragmentation RMCSA in Multi-core Fiber-based EONs” (Manuscript Number: photonics-2251162). The paper has been improved a lot. We have revised the manuscript according to the recommendations and have provided detailed responses to each comment below. For the reviewers’ convenience, we have highlighted major changes in the revised manuscript in yellow background.

Reply to Reviewer's comments:

Reviewer 3:

The authors depart from the fact that in multi-core elastic optical networks, minimizing inter-core crosstalk implies not to use the same spectral band as a communication channel in one of two, spatially close neighboring cores; hence, the idle channel; i.e., the non-used spectral band, remains as a fragmentation until the neighbor core is not using it. In this sense, not all the spectral bands or wavelength-channels can be used at the same time in all cores. This is particularly true between cores that are spatially close.

Re: Thank you very much for your valuable suggestion.

Your question is very correct. Based on the point of actual engineering, you believe that the method of minimizing cross-talk between cores is not using the same FSs of adjacent cores. The disadvantage of this allocation method is that it reduces the utilization rate of spectrum, which is viable at present when network capacity can meet the capacity demand. However, academic research should consider problems from a standpoint of leading edge, with a certain predictive nature. As the network demand is growing exponentially, the network capacity is constantly insufficient, so it is urgent to improve the efficiency of spectrum to transmit more services. Our research can improve the spectrum utilization rate under the precondition of satisfying the crosstalk constraint, which will not damage the service and can transmit more services.

The authors developed an algorithm, for the first time as they claim, that administrates the spectral-band assignation for a certain communication channel in such a way that the cross-talk is avoided (as much as possible) and the spectral fragmentation minimized; as a consequence, most of the spectral channels might be used at the same time in a multicore communication system.

Their report is well organized, the state of the art has been sufficiently revised and exposed, the concepts are clear; the hypotheses, formulas, programming and simulations make sense; the results compared to other works are clearly established and their simulations in quasi real environments are quite convincing.

In my opinion, the work is worth to be published in Photonics.

My general comments on the contents of the manuscript are:

Line 38: Please re-write “SDM-EONs based on MCF combined with space division multiplexing (SDM) technology of multi-core fiber (MCF)…crisis [4,5].”

Re: Thank you so much for your careful check. We rewrite this passage on line 38 as:

Space division multiplexing (SDM) based on multi-core fiber (MCF) can effectively expand optical fiber capacity by utilizing spatial dimension [4]. Thus, SDM-EONs, which combine MCF-based SDM and EONs, have become one of the effective solutions to overcome the capacity crisis [5].

Line 107. “Selva et al. in [17]” is not valid as such paper in not in references.

Re: Thank you so much for your careful check. We mention that the error in reference 17 is found by checking and have been modified

Line 234: donates or denotes? {please revise}

Re: Thank you so much for your careful check. We modify donates to denotes.

Figure 2.3: What is the meaning of the red colored Frequency slots? Please explain within the text as they are somehow immune to the considerations.

Re: Thank you so much for your careful check. The red colored resources represent the fragmentation. These resources are idle but not available because they are treated as protection bandwidth. We added the description in 3.2.2. Frequency Domain:

The red colored FSs represent the idle but unavailable spectrum resources known as fragmentation.

Lines 298-299: please clarify the word radio (ratio?)

Re: Thank you so much for your careful check. We modify radio to ratio.

Lines 356-357: please re-write the phrase “The time complexity in the CC-SCCF algorithm is consisted of the CC algorithm and the SCCF algorithm.”

Re: Thank you so much for your careful check. We rewrite this passage on line 38 as:

The time complexity in the CC-SCCF algorithm consists of the CC algorithm and the SCCF algorithm.

Round 2

Reviewer 1 Report

The authors have addressed the comments relatively well. I do not have any further questions.